# Revisiting Two Decades of Research Focused on Targeting APE1 for Cancer Therapy: The Pros and Cons

**DOI:** 10.3390/cells12141895

**Published:** 2023-07-20

**Authors:** Matilde Clarissa Malfatti, Alessia Bellina, Giulia Antoniali, Gianluca Tell

**Affiliations:** Laboratory of Molecular Biology and DNA Repair, Department of Medicine, University of Udine, 33100 Udine, Italy; matilde.malfatti@uniud.it (M.C.M.); bellina.alessia@spes.uniud.it (A.B.); giulia.antoniali@uniud.it (G.A.)

**Keywords:** APE1, inhibitors, cancer

## Abstract

APE1 is an essential endodeoxyribonuclease of the base excision repair pathway that maintains genome stability. It was identified as a pivotal factor favoring tumor progression and chemoresistance through the control of gene expression by a redox-based mechanism. APE1 is overexpressed and serum-secreted in different cancers, representing a prognostic and predictive factor and a promising non-invasive biomarker. Strategies directly targeting APE1 functions led to the identification of inhibitors showing potential therapeutic value, some of which are currently in clinical trials. Interestingly, evidence indicates novel roles of APE1 in RNA metabolism that are still not fully understood, including its activity in processing damaged RNA in chemoresistant phenotypes, regulating onco-miRNA maturation, and oxidized RNA decay. Recent data point out a control role for APE1 in the expression and sorting of onco-miRNAs within secreted extracellular vesicles. This review is focused on giving a portrait of the pros and cons of the last two decades of research aiming at the identification of inhibitors of the redox or DNA-repair functions of APE1 for the definition of novel targeted therapies for cancer. We will discuss the new perspectives in cancer therapy emerging from the unexpected finding of the APE1 role in miRNA processing for personalized therapy.

## 1. A Brief Introduction to APE1 Biology and Different Functions

The acronym APE1/Ref1 (or more simply APE1) stands for apurinic/apyrimidinic endodeoxyribonuclease-reduction/oxidation factor 1, which is a well-known protein with multifunctional roles ranging from the endodeoxyribonuclease activity on DNA and RNA to the hub role in several reduction/oxidation (redox) signaling pathways [1,2,3]. 

Historically, APE1 has been largely known for its function during the base excision repair (BER) pathway [4], in which non-bulky DNA lesions are repaired. In the BER, APE1 functions as the main specific endodeoxyribonuclease able to cleave abasic sites (AP), which are generated spontaneously or by the action of specific glycosylases. The single strand break (SSB) generated by APE1 cleavage is then brought to complete repair by other downstream BER enzymes (i.e., Polβ, XRCC1, FEN1, and Ligase III). Recent data have demonstrated that SSBs are also sensed by APE1 to initiate 3′-5′ SSB end resection and to promote ATR/Chk1-mediated DNA damage response (DDR) activation [5]. Indeed, through its exonuclease activity, APE1 generates a short ssDNA gap that, via PCNA and APE2, becomes a longer stretch of ssDNA coated by RPA that leads to the assembly of the ATR/Chk1 DDR complex [5,6]. Another main cellular role of APE1 is to function as a redox hub for several transcription factors (TFs). Indeed, the reduction of some TFs (i.e., NF-κB, p53, Hif-1α, AP-1, Pax-5/8, etc.) by APE1 allows their activation and, consequently, the initiation of the transcription of specific genes (i.e., IL-8, SIRT-1, VEGF, etc.). An additional role of APE1 in transcriptional regulation is due to its capacity to stabilize G-quadruplex (G4), which are stable conformational structures in the G-rich DNA portion of certain human promoters [7,8,9]. Although, following oxidative stress, the newly formed 8-oxoguanine (8-oxoG) can stall transcription due to its destabilizing effect, there is evidence that its presence in some promoters may induce the formation of BER-stabilized G4 that enhances gene expression [10,11]. In this context, the binding of APE1 to G4 sequences promotes G4 folding. Moreover, the acetylation of APE1 (acAPE1) enhances its residence time on DNA and stabilizes G4 structures in cells [12]. In this way, APE1 facilitates transcription factor loading at the promoter, thus modulating gene expression [13]. 

Finally, some recently characterized APE1 functions, especially those involved in RNA metabolism [14,15,16,17,18], have drawn particular attention. Specifically, APE1 has been demonstrated to be able to cleave abasic RNA [15] and damaged ribonucleotides embedded in DNA [17,18], revealing it as an efficient endoribonuclease. Furthermore, only in the last few years, in concomitance with its novel and unsuspected involvement in the RNA metabolism, it has been described that APE1 can be present in subcellular condensates formed through liquid-liquid phase separation mechanisms (LLPS) [19,20,21] and can be secreted by tumoral cells through extracellular vesicles (EVs) [22]. 

These intriguing APE1 roles have been discovered in three decades of constant interest in this protein, delineating them in physiological and pathological contexts and making APE1 an attractive therapeutic target for several pathologies, including cancer [23]. However, after more than 20 years of attempts, targeting APE1 still represents an important challenge in cancer therapy. In this review, we will focus on the dysregulation of APE1 in cancer, and then we will describe well-known inhibitors of the main functions of APE1, paving the way for novel functions involved in chemoresistance and potentially used as new therapeutic targets.

## 2. APE1 and Cancer: A Focus on Polymorphisms and Tissue Expression

As previously discussed, being involved in such focal cellular processes, the dysregulation of APE1 has a great impact on pathologies like cancer, making it an attractive therapeutical target [2,24]. APE1 dysregulation is involved in tumor development at three different levels, as it may concern alterations to its genetic sequence, expression, or localization [2]. It must be clearly stated, however, that, up to now, there is not clear evidence for a driver or passenger function of either of the above-mentioned alterations in the tumorigenic processes.

Several studies pointed out the importance of single nucleotide polymorphisms (SNPs) on the APE1 gene in cancer pathology [25] (Figure 1). The most common and studied APE1 variant is Asp148Glu (D148E), which is present in about 48% of the population [26]. X-ray crystallography experiments showed that this variant lacks significant structural changes and is considered benign [27]. As a matter of fact, the protein bearing this SNP holds normal AP endonuclease and DNA binding properties, but its 3′-RNA phosphatase and endoribonuclease activities are affected [16,28]. The role of this polymorphism in cancer is controversial due to several conflicting studies in the literature [26]. This common variant has been widely studied in more than one hundred publications. Indeed, numerous studies and meta-analyses observed an association between the D148E variant and an increased risk of different cancers, while others reported the opposite pattern even in the same tumor type [2,26,29].

Another biochemically studied APE1 polymorphic variant associated with cancer development is Arg237Cys (R237C) [30,31]. This substitution, which is prevalently observed in endometrial cancer [30,31], affects the functional activity of the whole protein [27]. X-ray crystal protein structure analysis revealed that this aminoacidic variation caused significant shifts in adjacent DNA binding residues, leading to a great decrease (~3-fold) of APE1 DNA binding ability [27]. Remarkably, this polymorphic variant showed a reduced ability to interact with its BER partners, such as Pol β and XRCC1 [25]. For example, X-ray data highlighted that the close lysine 244 (K244), which is implicated in the APE1-Pol β interaction [32], was shifted in the 3D structure, affecting the protein-protein interaction [27]. Moreover, the R237C variant showed reduced AP-endonuclease [25], 3′->5′ exonuclease, and 3′-damage excision [31] activities other than a reduced incision capacity close to nucleosomes [33]. 

One additional endometrial tumor-associated variant is Pro112Leu (P112L) [30], which exhibits comparable AP-endonuclease activity to the wild-type form [31]. 

In this review, we performed an analysis on cBioPortal to find SNPs or insertion/deletions (IN/DELs) in the APEX1 gene that are in association with different cancer types, considering a curated selection of non-redundant studies (213 studies selected, 69,223 samples, and 65,853 patients) (https://bit.ly/3M9Oya7 (accessed on 22 March 2023)) [34,35]. The somatic mutation frequency of APEX1 was 0.2%, with 108 unique variants. None of the variants detected represented a driver mutation for cancer, and most of them were sporadic. The most frequent variants were R193C/H, D148E, and H289Y/Q. Interestingly, even though R193C and R193H variants were detected to a comparable extent to D148E in the selected cohort of studies, there are no published works that focus on this mutation and its functional impact. According to the Mutation Assessor tool [36,37], R193H had a low impact on the protein’s functional activity, while R193C might have a worse one. There were no functional studies, even regarding H289Y/Q variants, which were predicted to have a neutral impact on protein activity. In Table 1, we report each variant found by using the cBioPortal tool, divided by tumor type. 

APE1 overexpression, as well as its altered localization, are prominent features of several different tumors, often with poor prognosis and malignant phenotypes. We hereafter provide a description of how APE1 is altered in cancer and how it impacts tumorigenesis (Table 2). 

In bladder cancer (BCa), several studies detected high expression levels of the APE1 protein in tumor tissues, compared to normal adjacent tissues, that were associated with poor outcomes [39,41,44]. APE1 overexpression was also linked to lymphovascular invasion features, as high VEGFA levels and an infiltration of CD163^+^ tumor-associated macrophages (TAMs) [42]. Moreover, the cellular distribution of APE1 was variable between high- and low-grade tumors. Whereas low-grade cancers displayed increased APE1 levels only in the nucleus, high-grade invasive tumors showed increased positive staining even in the cytoplasm [39]. APE1 is also a promising diagnostic biomarker in BCa, as its levels were increased in serum and urine when compared to normal healthy controls and were associated with tumor grade and stage, recurrence, and invasion [38,40,43]. Interestingly, a study observed an increased secretion of APE1 in bladder tumors displaying the D148E variant compared to the ones expressing the wild-type form, which contributed to increased serum levels of the protein in patients [129]. 

Concerning hepatocellular carcinoma (HCC), APE1 was upregulated at both transcriptional and translational levels compared to normal liver tissues [81,82,83]. Moreover, the mRNA content increased with tumor progression and was higher in less differentiated and more aggressive tumors [81]. Patients with higher APE1 protein levels exhibited unfavorable prognoses and a lower OS [82,83]. Interestingly, both APE1 truncated forms, missing the first 33 residues (N∆33–35 kDa), and APE1 full length (37 kDa), were detected in HCC tissue samples and HCC cell lines [84]. Moreover, the cellular distribution of APE1 was altered, with nuclear staining only in normal liver tissues while presenting a significant fraction of cytoplasmic positivity in tumor tissues [85]. Cytoplasmic APE1 was about three times higher in poorly differentiated tumors and was associated with a reduced OS [85]. Noteworthy, the cytoplasmic staining was prevalently associated with APE1 mitochondrial accumulation in grade 1 and grade 2 HCC tumors, but not in grade 3 tumors [86]. Even in HCC, APE1 serum levels can be exploited as a novel diagnostic biomarker, correlating with its overexpression in HCC tissues [84]. 

The APE1 protein was also overexpressed in pancreatic adenocarcinoma (PDAC) tissues and cell lines and associated with tumor aggressiveness and poor survival [122,123,124]. The proteolytic form of APE1 N∆33 has been detected even in PDAC tissues, with different abundances *versus* adjacent non-tumor tissue [55]. Interestingly, acAPE1 was overexpressed in PDAC tumors, while being almost undetectable in healthy pancreatic tissues [55,87]. acAPE1 has increased AP-endonuclease activity, which has been proposed as a cancer mechanism to overcome chemotherapy genotoxic stress and uphold proliferation [87]. APE1 localization in PDAC was mainly nuclear and similar between primary tumors and metastases [123]. An increased cytosolic localization was observed only in advanced tumor stages and was always concurrent with nuclear localization, while the complete absence of the cytoplasmic fraction was associated with invasion and poor differentiation [101,123]. 

Concerning prostate cancer (PCa), APE1 protein levels were upregulated compared to normal or benign hypertrophy (BPH) tissue [125,126]. Moreover, higher APE1 levels were observed in tumors bearing the TMPRSS2:ERG fusion [126]. APE1 localization was only nuclear in normal prostate tissue and non-cancerous prostate cell lines, while there was an increased expression in the cytoplasm compartment in tumor tissues and tumoral cell lines [125]. 

APE1 overexpression occurs also in oesophageal carcinomas, like oesophageal adenocarcinoma (EAC) [102,103,104] and oesophageal squamous cell carcinoma (ESCC) [105,106], probably as a mechanism adopted by cancer cells to survive the genotoxic effects of bile reflux [105,106]. APE1 localization was mainly nuclear and also associated with a worse OS in patients receiving platinum chemotherapy [101]. 

An in silico analysis identified APE1 as a central hub gene for gastric cancer, as its overexpression had a great prognostic value in two analyzed datasets (GSE1611533 and GSE54129) [64]. Indeed, APE1 is overexpressed at both transcriptional and translational levels in gastric cancer [65,66]. APE1 staining was weak in normal, non-cancerous gastric tissues, while it was high in tumor tissues. APE1 localization was detected both in the nuclear and cytoplasmic compartments of tumor tissues [66]. High levels of APE1 were also correlated with invasion and poor prognosis [66], as its serum levels are a valuable diagnostic biomarker for lymph node metastasis prediction [67]. 

APE1 was also upregulated in salivary gland carcinomas, and its levels increased depending on the malignant transformation process of the tumor [127]. APE1 overexpression was higher in smaller tumors displaying lymph node metastasis and invasive growth [127,128]. APE1 localization was mainly nuclear in every salivary gland tumor subtype analysed, except for adenoid cystic carcinomas, in which it was highlighted to present both nuclear and cytoplasmic localization [127,128]. 

Furthermore, the overexpression of APE1 protein and mRNA levels was also reported in non-small-cell lung cancers (NSCLCs) [88,89,90,91]. High APE1 expression was associated with poor prognosis, invasion, and chemoresistance as its levels increased upon treatment with platinum compounds [90]. Moreover, high APE1 serum levels, post-treatment, were correlated with a poorer OS [92]. Nuclear APE1 staining was associated with favourable patient outcomes [93], while a higher cytoplasmic localization was correlated with both poor survival and a shorter RFS [94,95,96]. Although both full-length and truncated forms were found in lung cancer, APE1 was prevalently truncated at the N-terminus in adjacent non-tumor tissues in NSCLC [55]. Moreover, acAPE1 was overexpressed in NSCLC tumors, with a strictly nuclear localization [55,87]. 

Several studies identified the overexpression of the APE1 protein in ovarian cancers, which has been associated with advanced tumor stages and decreased OS [101,114,115]. Moreover, patients with high levels of APE1 showed more frequent resistance to platinum therapy [101,114,116]. The interaction between APE1 and nucleophosmin 1 (NPM1) has been extensively examined in ovarian cancer, as the levels of the two proteins were positively associated with tumor aggressiveness, malignant phenotype, lymph node metastasis, and poor chemosensitivity [114,117]. It has been shown that compounds that impair this interaction can exert a synergistic effect on traditional chemotherapeutic molecules [118]. APE1 localization seemed to be heterogeneous in ovarian cancers, depending on the stage and histological subtype [24]. Some studies showed prominent cytoplasmic staining, which increased from well- to poorly-differentiated cancers and was higher in advanced-stage tumors [116,117,119,120]. In non-responding cisplatin patients, the observed APE1 overexpression was mainly at the cytoplasmic level, a feature that was also observed in cisplatin-resistant cell lines [116]. Interestingly, almost 90% of patients with abnormal levels of cytosolic APE1 displayed an abnormal distribution of NPM1 too [117]. Additionally, cytosolic APE1 can be considered an independent predictive factor for poor PFS and OS in ovarian cancer [119]. Other works showed prominent nuclear APE1 staining, which increased during tumorigenesis and was associated with survival time [115,121]. Additional studies showed an increase in APE1 in both compartments, but higher nuclear staining was associated with cancer aggressiveness, lower debulking after surgery, platinum resistance, and lower OS [101,116]. 

Concerning breast cancer, different studies reported conflicting results about APE1 protein expression levels. Some works described APE1 overexpression as mostly nuclear and associated with malignant phenotypes and an unfavourable prognosis [45,46,47]. Contrary to the pattern of increased acetylation observed in other cancer types [55,87], APE1 acetylation was lower in breast cancer compared to healthy tissues [48]. Even in this case, the functional interaction between APE1 and NPM1 in promoting platinum resistance has been described [49]. In contrast to these findings, another study showed that lower levels of APE1 were associated with tumor aggressiveness and a triple-negative phenotype [50]. Interestingly, in the Ki-67 low-level expression group, lower levels of APE1 were associated with poor OS [46]. 

APE1 protein levels were upregulated even in cervical tumors and were associated with Epithelial-to-Mesenchymal transition (EMT), lymph node metastasis, and poor radio-sensitivity [51,52,53,54]. APE1 localization was widely heterogeneous among cervical tumors, although with a main nuclear stain [52]. Remarkably, there was a significant difference in the subcellular localization of APE1 between radiotherapy non-responding and responding tumor cell lines. Indeed, radio-resistant cervical tumor cell lines showed higher levels of the cytoplasmic fraction and lower levels in the nucleus, suggesting a role for cytosolic APE1 in radio-resistance promotion [53]. 

Several studies described an overexpression of APE1 in colorectal cancers (CRC), observing a gradual increase in its expression during tumor progression [56,57,58,59] and in liver metastasis [60]. APE1 localization was heterogeneous between CRC cells, as the protein was found concurrently both in the nucleolus and cytoplasmic compartment, or, otherwise, it displayed an exclusive cytoplasmic localization [56]. Even in CRC tumor samples and cell lines, nuclear acAPE1 was overexpressed [55,61] and positively correlated with resistance to 5-Fluorouracil (5-FU) [61]. Moreover, both full-length and truncated forms were detected in colon cancer [55]. Interestingly, the levels of serum APE1 autoantibodies are valuable as diagnostic biomarkers for CRC [62]. 

Regarding gliomas, conflicting data are available. Some studies described an overexpression of APE1 in tumoral tissues compared to healthy ones [68,69], with a 13-fold increase in AP-endonuclease activity in 93% of tumors [68]. Glioma radioresistant cell lines displayed higher levels of APE1 compared to responding cell lines [70]. Indeed, an increase in APE1 expression was observed in patients after treatment and recurrence [71]. On the other hand, different studies have evidenced low mRNA and protein expression in adult high-grade gliomas, which was associated with poor OS [72,73]. Moreover, APE1 localization was predominantly nuclear [72].

Concerning melanoma, several studies identified APE1 overexpression at both transcriptional and translational levels [97,98,99]. Indeed, APE1 was overexpressed in melanoma cancer cell lines and in clinical samples, showing a prominent nuclear localization in both cases [98,99]. High mRNA levels were associated with vascular invasion, high proliferation rates, poor RFS, and OS [97,100]. Patients with higher levels of APE1 also showed a lower response to therapy [100]. APE1 was also overexpressed in another skin tumor, namely cutaneous squamous cell carcinoma (cSCC) [63], which was associated with increased proliferation and migration by EMT [63]. 

APE1 was dysregulated in several head and neck squamous cell carcinomas (HNSCC). In oral SCC (oSCC), APE1 was overexpressed at the protein level, and its high expression was significantly correlated with nodal status, shorter OS, and DFS [74,75]. APE1 localization was mainly nuclear, but translocation to the cytoplasm was observed after cisplatin treatment [74,76]. Moreover, APE1 serum levels represent a promising diagnostic biomarker [77]. Indeed, high levels of serum APE1 (sAPE1) were associated with late TNM stages, lymph node metastasis, and worse pathological differentiation [77]. Patients with lower levels of sAPE1 went through longer DFS after post-surgery cisplatin therapy and longer OS [77]. APE1 overexpression was also observed in laryngeal SCC (LSCC) [78], in sino-nasal SCC (sSCC), and in SCC with inverted papilloma (SCCwIP), with a vivid nuclear localization associated with metastatization [79]. Moreover, sSCC tumors showed higher cytoplasmatic staining compared to SCCwIP [79], which was associated with a higher T-stage and histological grade [79]. Lastly, APE1 overexpression, also characterized by lip SCC (lSCC), showed strong nuclear localization [80]. 

Furthermore, APE1 levels were upregulated in osteosarcoma and associated with poor prognosis and cisplatin resistance [107,108,109,110,111,112,113]. APE1 localization was both nuclear and cytoplasmic [107,112]. Patients with higher levels of the protein in the cytoplasmic content were less responsive to cisplatin treatment and experienced recurrence and metastasis [107]. 

Therefore, in general, APE1 is significantly overexpressed in different kinds of cancers, and subcellular distribution may significantly change depending on the specific tissue and tumoral stage, but in which way the overexpression and localization of APE1 in tumors are causally responsible for cancer onset and development, aggressiveness, and invasion is still debated. Currently, knowledge about the role played by APE1 polymorphic variants in cancer onset and progression is still unknown, as are the possible driver or passenger functions of APE1 mutations in cancer tumorigenesis. As mentioned above, the most accepted hypothesis regards the increased expression of APE1 in tumoral cells as they acquire a proliferative and chemoresistant phenotype. Several studies have proposed that the upregulation of APE1, as well as that of other BER enzymes, may underlie pro-survival mechanisms adopted by tumors to efficiently repair DNA damage, thus contributing to the onset of resistance mechanisms. Although the main function of APE1 is attributable to its endoribonuclease activity, it is believed that APE1 overexpression may also contribute to tumorigenesis through increased activity as a redox activator of several TFs, such as NF-κB, thus leading to an increase in tumor proliferation and survival and affecting the tumor microenvironment. We do believe that additional dysregulated functions of APE1, including dysregulation of RNA and miRNA metabolism and regulation of G4-structures containing promoter genes, could play an essential role in cancer development, although more detailed investigations are needed along these lines. 

In conclusion, further analysis on how and why altered APE1 expression is differentially associated with cancer development and metastasis should be a central aim of further study in the future. 

## 3. APE1 as a Still Promising Therapeutic Target after 20 Years of Research

In the last decades, APE1 has emerged as a promising therapeutic target in cancer, either for its role in DNA repair or in redox regulation of TF activities. In the next paragraphs, we will dive deeper into these functions of APE1, highlighting the study progression around the discovery of specific inhibitors, principally employed in chemotherapy (Table 3). Finally, we will discuss the new roles of APE1 in RNA metabolism and in cell signaling through its secretion, hypothesizing these novel functions as promising new targets in cancer therapy. 

### 3.1. Targeting the APE1 Endonuclease Activity

The endonuclease function of APE1, which is essential in the BER pathway, depends on residues sited on the C-terminal region of APE1. The most important amino acids involved in this activity are E96, which is implied in the coordination of divalent metals, and D210 and H309, both required for the hydrolytic reaction [130] (Figure 1). Other important residues that mediate different cleavage functions can be found in the C-terminal region too, including: D70, involved in the 3′-phosphodiesterase activity [131], K98, required in the Nucleotide Incision Repair (NIR) [132], and F266, implicated in the 3′-5′ exonuclease activity [133]. 

Previous studies identified different compounds that inhibit the endonuclease APE1 activity *in vitro* and in human cells, as summarized in different reviews [134,135,136,137]. Over the years, various groups have extensively worked towards the identification of specific small-molecule inhibitors able to target the DNA repair function of APE1 in combination studies, with the rationale that the blockade of APE1 endonuclease activity might have various therapeutic applications, particularly in cancer treatment, by sensitizing cancer cells to DNA-damaging agents and leading to tumor cell death. Although many studies support the inhibition of APE1 as a means of complementing current chemotherapeutic regimens and, accordingly, various chemical inhibitors have been developed, a clinical candidate has yet to be realized. Indeed, despite their high activity *in vitro*, the toxicity and selectivity in cells of the majority of the reported endonuclease inhibitors remain to be established. In this section, we attempt to present the major APE1 inhibitors identified thus far and discuss their activity. It is not within our scope to revisit all the inhibitors in depth; a comprehensive list of APE1 endonuclease inhibitors is reviewed in [130,137,138]. The published approaches utilized for the development of APE1 endonuclease inhibitors can be mainly categorized into: (i) screening of commercially available compounds that were synthesized for targeting other molecules; (ii) computational screening, and (iii) pharmacophore modeling. 

One of the first studied molecules impairing APE1 repair activity was Methoxyamine (MX), an alkoxyamine derivative that reacts to form an imine with the aldehyde group in the ring-open form of the abasic lesion, thereby indirectly blocking APE1 endonuclease activity [139,140]. Since MX was demonstrated to enhance the cytotoxicity effect of alkylating agents such as temozolomide (TMZ) in a wide variety of cancer cell lines both *in vitro* and in xenograft models [141,142], it advanced to clinical trials; however, to date, clinical studies have not shown any clear success. 

A second compound that was first identified as a radio-sensitizer of HeLa cells [143] and subsequently reported to be an inhibitor of APE1 by Luo and Kelly in 2004 [144] is Lucanthone, or Miracil D. Lucanthone was shown to enhance the cell-killing effect of TMZ and an alkylating agent such as methyl methanesulfonate (MMS) in culture and was further characterized by Naidu et al. to bind to the hydrophobic pocket site of APE1 [145]. No other studies were successively reported, but its specificity is still debated since a part of its inhibitory effect is mediated by its ability to intercalate within DNA and through the inhibition of topoisomerase II and possibly other cellular proteins [23].

Afterward, numerous laboratories relied on high-throughput screens (HTS), mainly based on fluorescence assays, to identify inhibitors of APE1 endonuclease activity. In general, the identification of the potential hits was followed by different assays aiming to prove specificity and selectivity for APE1 inhibition, including the AP site incision assay, the ability of the compound to bind DNA *per se*, and the ability to enhance the cytotoxicity of alkylating agents (i.e., TMZ, MMS). It is worth mentioning that the inhibitors reported so far showed affinities in the µM range that are not compatible with a suitable pharmaceutical agent, and more importantly, none of them has been demonstrated to have utility in pre-clinical animal cancer models. CRT0044876 (7-nitroindole- 2-carboxylic acid) is the first biochemically and biologically reported APE1 inhibitor identified through a fluorescein/dabcyl-based AP site incision assay [146]. Madhusudan et al. identified the compound CRT0044876 from a screening of a collection of structurally diverse small molecules. Despite the initial promising results obtained in the potentiation of the cell-killing effects of MMS and TMZ, the reproducibility of this compound has been brought into question [23,147], and because of its poor solubility and permeability, this compound has been further neglected until now, since its usage has been proposed conjugated with platinum [148]. Considering the weak results obtained with CRT0044876, other screenings were performed. Using a similar HTS approach, Seiple et al. screened the 2000-compound NCI Diversity Set of small molecules and identified aromatic nitroso, carboxylate, sulfonamide, and arylstibonic acid compounds with µM affinities for the APE1 protein [149]. Again, for these compounds, the relatively high inhibitory potency observed *in vitro* did not match a significant parallel effect in cells. Successively, various one-off studies did not progress to lead optimization. For example, in 2009, prompted by the evidence that APE1 represents an attractive therapeutic target in anticancer drug development, Zawahir et al. utilized a pharmacophore-based approach that was used to carry out a virtual screen of a 365,000 small molecule library [150]. The known interactions of APE1 with AP site-containing DNA, including components of hydrophobicity, H-bond acceptor, and negatively ionizable features, were utilized to design a virtual screen. In the same year, Simeonov et al. employing a quantitative HTS, screened the commercially available Library of Pharmacologically Active Compounds (LOPAC), identifying 6-hydroxy-DL-DOPA, Reactive blue 2, and myricetin as possible APE1 inhibitors [147]. Although these approaches predicted several potentially positive hits, they were not all tested in cell-based assays and thus have not been evaluated for cell permeability.

Successively, Kelley’s group has also used a fluorescence-based high-throughput assay to screen a library of 60,000 small-molecule compounds for their ability to inhibit the AP endonuclease activity of APE1 [151]. The most promising compounds were designated as APE1 Repair Inhibitor AR01, 02, 03, and 06. AR03 is chemically distinct from the previously reported small-molecule inhibitors of APE1. This compound was demonstrated to inhibit the cleavage of AP sites *in vitro* using whole cell extracts and to potentiate the cytotoxicity of TMZ and MMS in glioblastoma SF767 cells. Furthermore, very recently, AR03 was demonstrated to inhibit the exonuclease activity of APE1 in the SSB-induced ATR-Chk1 DDR pathway in human bone osteosarcoma U2OS cells, MDA-MB-231, and PANC1 [5,20]. While it is cell-permeable, its planar fused-ring structure may suggest its DNA intercalating ability, thus potentially being non-specific.

In 2011, Mohammed et al. focused on developing APE1 inhibitors for melanoma and glioma treatments using a structure-based drug design approach [98]. The crystal structure of APE1 was utilized to create four pharmacophore models, including the interactions of the previously identified inhibitor CRT0044876 with active site residues and molecular scaffolds designed to fit the ligand binding site. From the screening of 1679 hits, the authors identified compound 4 (N-(4-fluorophenyl)-2-(4-phenylsulfonyl-2- (p-tolyl)oxazol-5-yl) sulfanyl-acetamide) as the one with the highest AP endonuclease inhibitory activity and the potential to sensitize the activity of MMS and TMZ in both glioma and melanoma cell lines but not in HUVEC cells, suggesting specificity for malignant tissue. 

In 2012, a new class of inhibitors of the catalytic endonuclease function was identified by Aiello et al. [152]. Compounds 32–35, which have a 3-benzylcarbamoyl-2-methoxybenzoic acid structure, showed the most active and selective inhibition activity of APE1. These compounds have the potential to be used in combination therapy with 5-fluorodeoxyuridine for colon cancer treatment. In the same year, using docking-based virtual screening, 15 potential compounds were identified as inhibitors of APE1 from a library of over 4 million molecules [153]. Two of these compounds, 36 and 37, were found to be potent inhibitors of the protein and could increase the toxicity of MMS. Through molecular dynamics simulations, it was discovered that these compounds may interact with the protein through important binding modes such as hydrogen bonds with specific residues and hydrophobic interactions by virtue of their quinoxaline core. In 2012, Simeonov’s group performed a fully automated HTS using a kinetic fluorescence assay on the NIH Molecular Libraries Small Molecule Repository and other collections, examining each agent at different concentrations [154]. They identified active APE1 inhibitors able to potentiate the genotoxic effect of MMS, leading to an increase in AP sites. The chemical structures of the most effective inhibitors, namely MLS001196838, MLS000587064, MLS000737267, MLS000090966, and MLS000863573, would have served as starting points for medicinal chemists to further optimize them.

Another fluorescence-based quantitative HTS of 352,489 small molecules from the NIH Molecular Libraries Small Molecule Repository was performed by Rai et al. [155]. APE Inhibitor III (N-(3-(1,3-Benzo[d]thiazol-2-yl)-6-isopropyl-4,5,6,7-tetrahydrothieno[2,3-c]pyridin-2-yl)acetamide) has been demonstrated to potentiate MMS and TMZ activity in HeLa cells. This compound was further used and distributed by the sellers as one of the most promising APE1 inhibitors for both its endonuclease and exonuclease activities; however, this compound has not significantly advanced beyond *in vitro* studies. Using the crystal structure of APE1, Srinivasan et al. computationally constructed molecules that would sterically block its endonuclease site and identified molecules that all contain the 2-methyl-4-amino-6,7-dioxoloquinoline structure [156]. The mechanism of action of the compounds was probed by fluorescence and competition studies in T98G glioma cell lines, which indicated for compounds 1 and 4 a direct interaction between the inhibitor and the active site of the APE1 protein.

In 2015, a pharmacophore model for APE1 small-molecule inhibitors was used to identify new compounds by means of in silico screening of 10,159 compounds [157]. The virtual docking assay identified four compounds with a 2-methyl-4-amino-6,7-dioxoloquinoline core (AJAY 1–4); AJAY 4 showed the best results in the inhibition of cell growth; however, none of the compounds have advanced in clinical studies.

Another novel in silico approach was pursued by Trilles et al., who, guided by X-ray crystal structures of APE1 and computational docking of solvents, identified binding hotspots for small organic molecules [158]. Accordingly, they screened a library of macrocycles for inhibition of APE1 endonuclease activity and identified four novel macrocycles that they used as a starting point for designing APE1 ligands. From the initial screening of 66 compounds, only four exhibited concentration-dependent inhibition of APE1 endonuclease activity (MC043, MC047, MC042, and MC019). Building on these hits, additional macrocycles were synthesized, and macrocyclic lactams 13, 21, and 24 have been demonstrated to be more effective in inhibiting APE1 endonuclease function in combination with MMS.

Unfortunately, none of the compounds developed so far have advanced to significant *in vivo* studies or clinical trials. Very recently, two works argued about the specificity of some of the most prominent compounds that are usually sold by suppliers as APE1 inhibitors. In the work of Pidugu et al., it has been demonstrated through structural, biophysical, and biochemical approaches that several reported small molecules are weak APE1 inhibitors [159]. In particular, through an NMR chemical shift perturbation assay, they showed that CRT0044876 and three similar indole-2-carboxylic acid compounds (5-fluoroindole-2-carboxylic acid [98], 5-nitroindole-2-carboxylic acid, and 6-bromoindole-2-carboxylic acid) bind at a pocket of APE1 that is distal from its active site. Furthermore, using Dynamic light scattering (DLS), they also demonstrated that CRT0044876 [146], myricetin [147], and APE Inhibitor III [155] form colloidal aggregates that could sequester APE1, causing non-specific inhibition. For this latter compound, Xue and Demple recently questioned about the specificity of this molecule [138]. Since APE1 knock-out lines (CH12F3 [160]) showed equal sensitivity to direct killing by APE Inhibitor III, being even more sensitive to APE Inhibitor III than its wildtype counterpart, the authors claimed possible off-target effects that must be taken into account when using these inhibitors at high dosages.

### 3.2. Targeting the APE1 Redox Activity

Unlike the BER function, which is highly conserved from prokaryotes (*E. coli* exonuclease III) to humans, the redox function is probably unique to mammalians [161]. Whereas the C-terminal of APE1 is mainly involved in the regulation of endodeoxyribonuclease activity, the N-terminal, principally consisting of an unstructured region, is strongly implicated in protein-protein interactions and in the activation of several TFs via a redox mechanism. Specifically, the redox function of APE1 is exploited by cysteine residues sited at positions 65, 93, and 99 of the N-terminal region (Figure 1). These residues are involved in the redox cycle responsible for controlling the reduced state of several TFs [161]. By reducing the TFs, APE1 makes them able to bind DNA. APE1 then returns to its basal state through another reduction that occurs via a thiol/sulfide exchange with thioredoxin. Among the several TFs regulated by APE1, we include the principals such as NF-κB [162], AP-1 [162], HIF1α [163], STAT3 [164,165], p53 [166], NRF2 [167], Pax-5 and -8 [168], and others [169]. Given the roles of all these TFs in cellular biological processes, the effects of APE1 as a redox signaling factor regard principally the promotion of growth, migration, DDR signaling, and survival in tumor cells, as well as inflammation and angiogenesis in the tumor microenvironment. Thus, inhibition of APE1 redox activity can be a target for slowing growth and progression during tumoral processes. Indeed, the pharmacological inhibition of APE1 redox activity causes a decrease in the ability of the TFs to bind to DNA [169,170,171] and thereby increases the cancer cells’ response to chemotherapeutic agents [172,173].

Differently from the AP-endonuclease inhibitors, testing redox inhibitors resulted in more complications during these years, due in part to the arduous modalities of detection of the redox activity of APE1. In this paragraph, we propose a roundup of the literature on a few redox inhibitors that have emerged on the scientific scene.

Dietary agents and several compounds from natural sources, such as soy isoflavones, resveratrol, and curcumin, as well as the vitamins ascorbate and α-tocopherol [174], were initially tested. Curcumin is a polyphenol with the potential for treatment or prevention of particular human diseases such as oxidative and inflammatory conditions, metabolic syndrome, arthritis, anxiety, hyperlipidemia, and cancer [175]. In 2017, it was demonstrated that curcumin affected the APE1 redox function, inhibiting the transcriptional activity of APE1 on AP-1 and NF-κB genes *in vitro* [176]. For its multiple anti-inflammatory, antioxidant, and anti-neoplastic properties, curcumin has been enrolled in more than 300 clinical trials. Resveratrol is a naturally occurring polyphenolic compound present in red wine and grapes. It has been demonstrated that it exhibits a neuroprotective role in models of central nervous system diseases, including cerebral ischemia/reperfusion injury [177]. By inhibiting APE1 redox function, resveratrol caused a significantly diminished activity of AP-1 and NF-κB proteins in different human cancer models, enhancing the cytotoxicity of chemotherapy [99]. Similarly, utilizing soy isoflavones to block redox signaling through APE1 and NF-κB dramatically increased prostate cancer cells’ sensitivity to radiation [178].

About ten years ago, the Kelley’ group synthesized a molecule that turned out to be highly promising in the inhibition of APE1 redox activity in several cancer models [24,179]. This molecule [(2E)-2-[(4,5-dimethoxy-2-methyl-3,6-dioxo-1,4-cyclohexadien-1-yl)methylene]-undecanoic acid, commonly denoted as APX3330 (or E3330), is a quinone derivative. Several studies were then performed using different pathological models both in vitro and in vivo, in which it was demonstrated that APX3330 selectively inhibited NF-κB-mediated gene expression through APE1 binding [180]. In 2009, Zou et al. demonstrated that APX3330 blocked the in vitro growth of pancreatic cancer-associated endothelial cells and the differentiation of bone marrow-derived mesenchymal stem cells into CD31(+) endothelial progeny. Specifically, the effect was attributable to a reduction of H-Ras expression and intracellular nitric oxide (NO) levels, as well as decreased DNA-binding activity of HIF-1α. Inhibition of the APE1 redox function by APX3330 might be a potent therapeutic strategy in solid tumors [181]. Indeed, APX3330 showed anticancer properties in pancreatic cancer, including inhibition of cancer cell growth and migration in several cancer cell lines and xenograft models in mice [182]. APX3330 inhibited the proliferation, migration, and tube formation of retinal vascular endothelial cells in vitro and reduced retinal angiomatous proliferation and neovascularization in vivo [183]. As anticipated in the previous paragraphs, elevated expression levels of APE1 have been correlated with more aggressive phenotypes and a poor prognosis for NSCLC. Recently, Manguinhas et al. demonstrated that APX3330, in combination with cisplatin, reduced H1975 cell viability, migration, and invasion, highlighting its use as a boost for cisplatin in NSCLC cells [184]. Moreover, the inhibition of APE1 redox function through APX3330 combined with docetaxel treatment decreased the proliferative rate, migration, and invasion of MDA-MB-231 breast cancer cells [185]. The APX3330 inhibitory activity was also assessed in pathological angiogenesis, such as retinal neovascularization [186]. Li et al. demonstrated that APX3330 treatment suppressed experimental choroidal neovascularization *in vitro* and *in vivo*, demonstrating that APE1 regulates multiple TFs and inflammatory molecules and is essential for CEC angiogenesis. That could represent a novel candidate for therapeutically targeting neovascular eye diseases and alleviating the burden associated with anti-VEGF intravitreal injections [187]. Recently, it has also been demonstrated that APX3330 has the potential to be used for the treatment of γ-herpesvirus infection and associated diseases [188].

Given its promising and potential anti-angiogenic and antineoplastic activities obtained *in vitro*, APX3330 was enrolled in the APX_CLN_0011 Phase 1 clinical trial in 2017. This trial (ClinicalTrials.gov (accessed on 4 April 2023) Identifier: NCT03375086) was a multi-center, open-label, dose-escalation oncology study of APX3330 in patients with advanced solid tumors. The study was completed in 2020, showing the assessment of APX3330’s safety, anti-tumor activity, pharmacokinetic and pharmacodynamic profile [189], and the recommendations for the Phase 2 study dose. Oral APX3330 demonstrated a favorable safety and tolerability profile and was suitable for Phase 2. In 2020, APX3330 was enrolled in the ZETA-1 Phase 2 clinical trial to evaluate its safety and efficacy to treat diabetic retinopathy and diabetic macular edema. The trial was completed this year (2023) (ClinicalTrials.gov (accessed on 4 April 2023) Identifier: NCT03375086). Oral administration of APX3330 and placebo has demonstrated a favorable ophthalmic and systemic safety and tolerability profile. Additional safety data from the ongoing ZETA-1 trial will be evaluated to further characterize the efficacy and safety of APX3330 for the oral treatment of diabetic eye diseases.

On the basis of the results obtained with APX3330, new analogues of this inhibitor were synthesized, including RN8–51, 10–52, and 7–60 [190]. Data have demonstrated that especially the analogue RN8–51 decreased cancer cell growth with little apoptosis, demonstrating itself as particularly promising for further anticancer therapeutic development. Kelley et al. synthesized novel, second-generation APE1 redox-targeted molecules such as APX2007, APX2009, APX2014, and APX2032 and determined whether they would be protective against neurotoxicity induced by cisplatin or oxaliplatin while not diminishing the platins’ antitumor effect. Specifically, they used an ex vivo model of sensory neurons in culture, through which they demonstrated that especially APX2009 [(2E)-2-[(3-methoxy-1,4-dioxo-1,4-dihydronaphthalen-2-yl)methylidene]-N,N-diethylpentanamide] was an effective small-molecule neuroprotective against cisplatin and oxaliplatin-induced toxicity. APX2009 also demonstrated a strong tumor cell killing effect in monodimensional cultured tumor cells, which was further substantiated in a more robust three-dimensional pancreatic tumor model [179]. Together, these data suggested that the second-generation compound APX2009 was effective in preventing or reversing platinum-induced CIPN while not affecting the anticancer activity of platins [179]. Moreover, all three compounds (APX2007, APX2009, and APX2032) demonstrated similar inhibition of NF-κB binding [179].

Finally, in 2010, Nyland et al. described a series of quinones, including benzoquinone and naphthoquinone, analogues of APX3330, with the ability to reduce tumor growth [191].

### 3.3. Targeting Both the APE1 Endonuclease and Redox Activities

Very few molecules have been demonstrated to inhibit both APE1 endonuclease and redox function. Among them, one was Gossypol [192]. Gossypol is a natural polyphenolic aldehyde that exhibits various effects, including antioxidant, anticancer, antiviral, antiparasitic, and antimicrobial activities. It can directly interact with APE1 and enhance the cell-killing effects of MMS and cisplatin. A recent clinical trial (ClinicalTrials.gov (accessed on 4 April 2023) Identifier: NCT00540722) aimed to investigate the potential clinical benefit of combining Gossypol with docetaxel and cisplatin in patients with NSCLC who have high expression of APE1 [193]. The trial, designed as a prospective and randomized study, did not show a significant difference between the Gossypol and placebo groups, although the Gossypol-treated patients had better outcomes in terms of increased PFS and OS.

One additional inhibitor was AT-101, a derivative of Gossypol and an oral inhibitor of the anti-apoptotic Bcl-2 and Bcl-xL proteins. AT-101 has been shown to exhibit potent anticancer activity, although its chemosensitizing effects are not fully understood. Indeed, AT-101 enhanced the sensitivity of A549 cells to cisplatin *in vitro* and *in vivo* by inhibiting APE1-mediated IL-6/STAT3 signaling activation, suggesting its potential use in NSCLC chemotherapy [194]. Moreover, it was also found to suppress gastric cancer cell migration and renewal and promote chemotherapeutic sensitivity in a gastric cancer model in vivo [195]. The molecular mechanism of its anticancer activity via inhibition of the endoribonuclease or redox activities of APE1 remains unclear.

## 4. Future Perspectives from Targeting the Non-Canonical Roles of APE1 in miRNA Processing

We recently proved that APE1 contributes to the expression of chemoresistance genes via functions in RNA metabolism involving miRNAs. We found that APE1: (i) binds to structured RNAs, including pri-miRNAs [14,196]; (ii) is involved in the processing of miRNAs implicated in cancer development (e.g., miR-221/222, miR-1246, miR-130b, miR-146a) [197]; and (iii) is a central hub connecting different subnetworks of cancer-associated proteins involved in RNA metabolism and miRNA sorting (e.g., NPM1, hnRNPA2/B1, AUF1, FUS, and SFPQ) [196,198,199]. We demonstrated that, during genotoxic stress, nuclear APE1 favors the processing and stability of miRNA precursors through its association with the DROSHA microprocessor complex, impacting, for example, the miR-221/222 axis and, in turn, modulating the expression of the tumor suppressor PTEN [14]. Using NSCLC cancer cell lines, we recently defined a signature of 13 miRNAs (miR-1246, miR-4488, miR-24, miR-183, miR-660, miR-130b, miR-543, miR-200c, miR-376c, miR-218, miR-146a, miR-92b, and miR-33a) that strongly correlate with APE1 expression in human lung cancer and play a central role in cancer cell proliferation and survival [197]. Whether these APE1-regulated miRNAs are responsible for cancer cell response to genotoxic treatment and explain the role of APE1 in chemoresistance through post-transcriptional mechanisms is still unknown and should be addressed to understand the central role of APE1 in cancer progression and to define new antitumor strategies. It should be defined whether APE1 recognizes specific oncogenic miRNAs alone or in combination with specific proteins through the detection of regulatory motifs present in the miRNA structure.

## 5. Secreted APE1 as a Novel Prognostic Non-Invasive Biomarker of Cancer Development

We recently showed that enzymatically active APE1 can be secreted (sAPE1) by cancer cells through EVs, including exosomes, during genotoxic stress conditions [8]. However, APE1’s presence in the extracellular milieu is still poorly characterized [186,187,188]. sAPE1 expression is actually considered a novel biomarker for the prognosis of NSCLC, as proved in a previous study performed on NSCLC patients, in which their levels of sAPE1 were significantly higher compared to healthy controls and were associated with a worse PFS [198]. Recently, data obtained by our research group confirmed these observations in a cohort of HCC patients [190], in which we found that sAPE1 levels correlated with poor prognosis and were able to discriminate between cancer patients and cirrhotic or healthy donors. The presence of this protein in the sera of patients is not solely restricted to cancer diseases but also in inflammatory models, such as coronary artery disease and endotoxemia [191,192]. The biological function of sAPE1 is still completely unknown. An intriguing hypothesis sees its action as a paracrine molecule triggering cell-to-cell communication, which is important for the local tissue microenvironment’s inflammatory response.

Evidence on the mechanisms responsible for APE1 secretion is lacking, even though the importance of the acetylation, occurring on specific lysine residues sited in the first 33 N-terminal portions of the protein (K27, K31, K32, and K35), has been highlighted in cells treated with the histone deacetylase inhibitor trichostatin A [14]. It seems reasonable that APE1 secretion might derive from EV formation via the endosomal sorting complex (ESCRT), due to the protein lacking a classic secretory signal peptide [193]. This pathway is responsible for the biogenesis and maturation of multivesicular bodies (MVBs), composed of many intraluminal vesicles (ILVs), that are released in the extracellular milieu as exosomes. ILV formation can occur through several mechanisms, and information about the regulation of these processes and the possible differences between the promoted cargo selections is still missing [194].

It is conceivable that these vesicles might be highly shuttled between cells within the tumor mass and deliver their content to target cells. This process may fulfill the cancer cells’ requirement for a high amount of APE1 to counteract the DNA damage inferred by drugs in a paracrine manner, suggesting that APE1-secretion could represent a novel damage-associated molecular pattern (DAMP) mechanism that deserves further in-depth study to develop inhibitors that could specifically target alterations of APE1 secretion in different cancers.

## 6. Conclusive Remarks and Future Perspectives

Despite the high potency of many of the compounds aforementioned, additional work is necessary to deliver more specific inhibitors of APE1-altered functions in tumors, which could be useful for clinical trials. While progress has certainly been made in identifying potent APE1 inhibitors, further efforts are needed to specifically achieve selectivity and efficacy. This will require consideration of both the abasic site binding pocket and more distal features of the enzyme that might be important for DNA binding. X-ray crystallography and *in vivo* experiments would be crucial to expedite rational inhibitor design, validate APE1 as a target, and explore possible side effects. Moreover, knowing the multiple APE1 cellular functions and their detailed molecular mechanisms could allow us to better target APE1 dysregulation in pathologies (Figure 2).

The discovery of novel functions for APE1 is constantly evolving. As mentioned in the introduction, the ability of APE1 to recognize and process SSBs through its 3′-5′ exonuclease activity [5,6] could represent an interesting target for developing new inhibitors specifically directed against this APE1 function and able to inhibit *in ultimum* the promotion of the ATR/Chk1-mediated DDR activation.

Moreover, targeting a protein-nucleic acid interaction is challenging, and this has contributed to the limited success in developing APE1 inhibitors. New approaches are needed for the discovery of novel and selective APE1 inhibitors. In this context, Wilson DM III et al. proposed the application of fragment- and structure-based drug discovery (FBDD/SBDD) methods in the quest for new clinical agents [200]. They applied the ABSOneStepTM platform, identifying 25 high-quality crystal structures showing unique and diverse fragment hits bound at the endonuclease site as well as at a previously unidentified secondary site, overall suggesting multiple novel strategies for inhibiting APE1. Indeed, in addition to direct inhibitors of APE1 nuclease activities, inhibitors against other functions of APE1 may also be clinically valuable. Considering the complex role of APE1, exploring allosteric modes of inhibition, such as disrupting vital interactions between APE1 and other cellular protein partners, might be an alternative option [118,201]. For example, we demonstrated that the molecular association with NPM1 modulates the endonuclease activity of APE1 [118]. HTS for the disruption of this interaction led to the discovery of three compounds (fiduxosin, spiclomazine, and SB 206553); of these, fiduxosin and spiclomazine displayed anti-proliferative activity and sensitized cells to bleomycin. A synergistic effect with platinum drugs was also observed by using these inhibitors in a triple-negative breast cancer cell model, demonstrating how APE1 could also represent a useful therapeutic biomarker in this type of tumor [49,202]. Similarly, the disruption of other APE1 protein interactions or functions can be taken into consideration.

**Table 3 cells-12-01895-t003:** List of the principal APE1 inhibitors. The principal APE1 inhibitors are grouped by the APE1 function inhibited, including the endonuclease activity, the redox activity, and the protein-protein interaction. For each inhibitor, the IUPAC name, the PubChem CID, the molecular formula and weight (MW), and the structure (obtained with PubChem Sketcher V2.4) are reported. At the end, the main references for each inhibitor are indicated. For more detailed information, refer to the text.

APE1 Function-Inhibited	Name	IUPAC Name	PubChem CID	Molecular Formula	MW (g/mol)	Structure	Refs
**Endonuclease**	**Methoxyamine**	O-methylhydroxylamine	4113	CH_5_NO	47.057	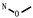	[139,140,141]
**Lucanthone**	1-[2-(diethylamino)ethylamino]-4-methylthioxanthen-9-one	10180	C_20_H_24_N_2_OS	340.5	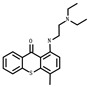	[143,144,145]
**CRT0044867**	7-Nitroindole-2-carboxylic acid	81409	C_9_H_6_N_2_O_4_	206.15	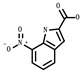	[23,146,147,148]
**Myricetin**	3,5,7-trihydroxy-2-(3,4,5-trihydroxyphenyl)chromen-4-one	5281672	C_15_H_10_O_8_	318.23	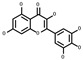	[147]
**AR03**	2,4,9-Trimethylbenzo[b][1,8]naphthyridin-5-amine	698490	C_15_H_15_N_3_	237.30	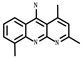	[151]
**APE Inhibitor III**	N-[3-(1,3-benzothiazol-2-yl)-6-isopropyl-4,5,6,7-tetrahydrothieno[2,3-c]pyridin-2-yl]acetamide	3581333	C_19_H_21_N_3_OS_2_	371.5	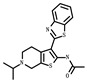	[155,156]
**Redox**	**Curcumin**	(1E,6E)-1,7-bis(4-hydroxy-3-methoxyphenyl)hepta-1,6-diene-3,5-dione	969516	C_21_H_20_O_6_	368.4	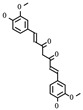	[175,176]
**Resveratrol**	5-[(E)-2-(4-hydroxyphenyl)ethenyl]benzene-1,3-diol	445154	C_14_H_12_O_3_	228.24	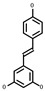	[99,177]
**APX3330**	(2E)-2-[(4,5-dimethoxy-2-methyl-3,6-dioxocyclohexa-1,4-dien-1-yl)methylidene]undecanoic acid	6439397	C_21_H_30_O_6_	378.5	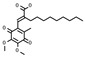	[24,179,180,181,182,183,184,185,186,187,188,189]
**APX2009**	(2E)-N,N-diethyl-2-[(3-methoxy-1,4-dioxonaphthalen-2-yl)methylidene]pentanamide	71618575	C_21_H_25_NO_4_	355.4	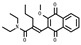	[179]
**Endonuclease Redox**	**Gossypol**	7-(8-formyl-1,6,7-trihydroxy-3-methyl-5-propan-2-ylnaphthalen-2-yl)-2,3,8-trihydroxy-6-methyl-4-propan-2-ylnaphthalene-1-carbaldehyde	3503	C_30_H_30_O_8_	518.6	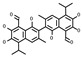	[192,193]
**Protein-protein interaction**	**Fiduxosin**	5-[4-[(3aR,9bR)-9-methoxy-3,3a,4,9b-tetrahydro-1H-chromeno[3,4-c]pyrrol-2-yl]butyl]-12-phenyl-8-thia-3,5,10,13-tetrazatricyclo[7.4.0.02,7]trideca-1(13),2(7),9,11-tetraene-4,6-dione	172307	C_30_H_29_N_5_O_4_S	555.6	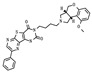	[49,118,202]
**Spiclomazine**	8-[3-(2-chlorophenothiazin-10-yl)propyl]-1-thia-4,8-diazaspiro[4.5]decan-3-one	65714	C_22_H_24_ClN_3_OS_2_	446.0	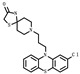	[49,118,202]
**SB 206553**	1-methyl-N-pyridin-3-yl-6,7-dihydropyrrolo[2,3-f]indole-5-carboxamide	5163	C_17_H_16_N_4_O	292.33	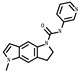	[49,118,202]

A new hot topic concerns RNA G-quadruplexes (RG4s), which are disease-associated non-canonical structures composed of stacks of guanine tetrads (called G-quartets) kept together by Hoogsteen hydrogen bonds. RG4s are increasingly recognized as fundamental post-transcriptional regulators of gene expression [203]. Interestingly, these elements are widespread in the transcriptome and are particularly enriched in miRNAs [204]. The folding of these structures can be controlled by their RBP interactors (i.e., hnRNPA2B1, FUS, etc.), cations (i.e., K^+^), and small molecule ligands [205], making RG4s highly dynamic. Very recent data underline a regulatory function played by RG4 in miRNA maturation through DROSHA- and Dicer-inhibition [206] and a potential role in physiological and pathological LLPS [207,208]. The presence of the RG4 structure in pre-miRNA exists in equilibrium with the canonical stem-loop structures, and this equilibrium regulates the maturation of some miRNAs, such as miR-92b [209]. However, mechanistic information on RG4 function in miRNA sorting is missing, as is information on the functional role of oxidized guanine (8-oxo) or abasic (AP) sites in the RG4-forming structures in the stability and biological properties of the miRNAs in which these structures are present. Understanding whether APE1 function in miRNA processing and degradation could be driven by RG4-mediated folding will open mechanistic views as well as translational applications in cancer biology. We are working along these lines.

Finally, this fascinating field of research relies on the findings that APE1 can be secreted in the extracellular milieu through EVs. Understanding the intracellular routes responsible for this secretion in cancer cells and the role of sAPE1 as a potential paracrine molecule will open new perspectives on precision medicine.

## Figures and Tables

**Figure 1 cells-12-01895-f001:**
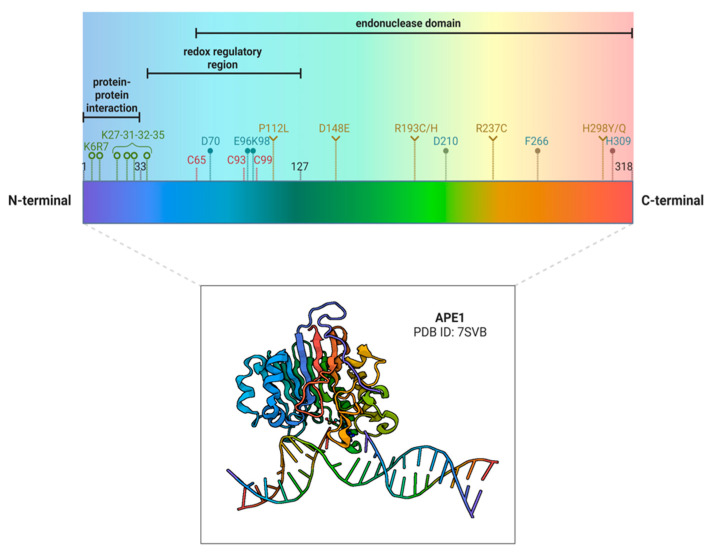
Human APE1 structure and its key residues. A schematic representation of the primary sequence of APE1, in which the most important aminoacids are highlighted. In the first 33 residues required for protein-protein interaction, residues K6 and K7 (green, ο) are involved in the shuttling between the nucleus and the cytoplasm. Residues K27, K31, K32, and K35 are essential for proteosomal cleavage. All the above mentioned lysine residues can also be acetylated. The redox regulatory region is included between aminoacids 35 and 125, in which the main residues involved are C65, C93, and C99 (red). The endonuclease domain spans between 65 and 318 (residues indicated in blue, •). Specifically, E96 is involved in divalent metal coordination, while D210 and H309 have functions in the hydrolytic reaction. Other important residues in the endonuclease domain are D70, which is implicated in the 3′-phosphodiesterase activity; K98, important for the nucleotide incision repair (NIR); and F266, which is involved in the 3′-5′ exonuclease activity. Lastly, P112L, D148E, R193C/H, R237C, and H298Y/Q (yellow, ο) are some of the polymorphisms of APE1 that will be discussed in this review. Below, the tridimensional structure of APE1 is reported (PDB ID: 7SVB). Created with BioRender.com (accessed on 31 May 2023).

**Figure 2 cells-12-01895-f002:**
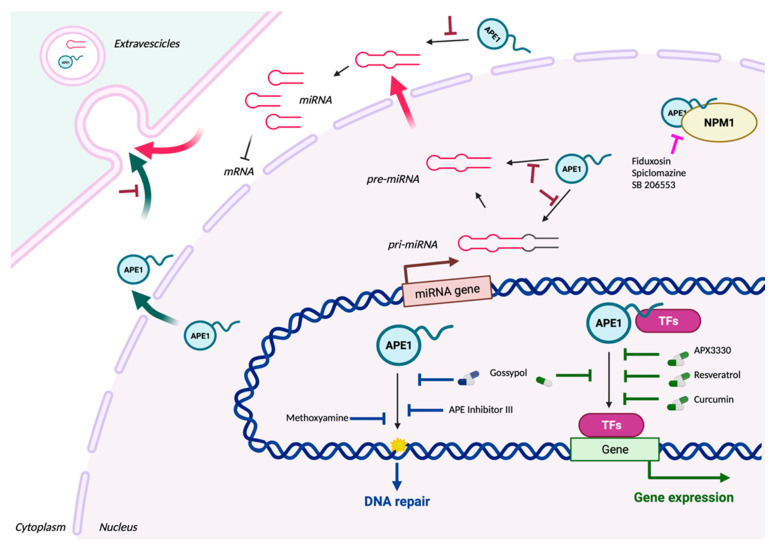
Illustration of the main functions of APE1 and its relative inhibitors. The DNA repair mediated by the endonuclease activity of APE1 is inhibited by Methoxyamine, APE Inhibitor III, and Gossypol (blue blunt arrows). The gene expression promoted by the redox activity of APE1 on TFs is inhibited by Gossypol, curcumin, resveratrol, and APX3330 (green blunt arrows). Especially through its N-terminal region, APE1 is involved in several PPIs, including Nucleophosmin 1 (NPM1). This interaction is inhibited by fiduxosin, SB 206553, and spiclomazine (fuchsia blunt arrows). Recent findings have shown that APE1 is secreted by EVs in the microambient and is involved in RNA metabolism, including the regulation of miRNA processing. If these novel features of APE1 can be inhibited, it is still unknown (brown blunt arrows) and an interesting starting point for future explorations. For each inhibitor, pillows are drawn when the inhibitor has been enrolled in one or more clinical trials. Created with BioRender.com (accessed on 31 May 2023).

**Table 1 cells-12-01895-t001:** APE1 polymorphisms detected in different cancer types by using the cBioPortal tool. * stands for type of mutation. Further information is available at https://bit.ly/3M9Oya7 (accessed on 22 March 2023).

Cancer	Polymorphisms
Adrenocortical carcinoma	D283H; X20_splice
Ampullary cancer	M271T
Bladder cancer	D15H; S146L; Q109E; E217 *
Bone cancer	V131M
Breast cancer	L244Tfs * 8; Q51 *; H289Y; G5A; K7Rfs * 75; R185W; A60G
Cervical cancer	R281C; K7R
Colorectal cancer	A273T; R193C; R221H; G130D; R247Q; A230D; M271I; R221H; R181 *; E242D; R274 *; P293S; P49Qfs * 33; L220I; T233M; N226Efs * 26; V131M; L291Vfs * 6; R221C; K63E; D210N; R281H; X82_splice; E101D; K77N; P331H; G306S; G132D
Endometrial cancer	R193H; S164L; R221H; A170V; P223L; V84I; V278D; R281H; K228T
Gastric cancer	R281C; P122A; K27N
Glioma	H289Q; Q245R; G80E; D142N
Head and Neck cancer	P48H
Leukemia and Lymphoma	R181Q; L17P; K7R; R187H
Liver cancer	G5E; W280S; L220P; W280R; D50Rfs * 28; G8R; H289Y, N226Efs * 26; Y264_G279del
Lung cancer	G41C; V206Cfs * 11; V142Sfs * 8; E16Q; P331T; E149Q; X147_splice; R177*; D90H; Q51 *; X237_splice; R193C; R28S; M271del; V206Cfs * 11; S115F; G8R; I146V; D148E
Melanoma	G241W; E16K; G127V; R136S; P122T; D148E; A263V; K7Rfs * 75; L108F; V69L
Oesophageal cancer	E46D; D251N; M270Nfs * 14; F165V; N102I; L291Vfs * 6; K3R; G145D
Ovarian cancer	Q95 *, V168I; R193H; L291Vfs * 6
Pancreatic cancer	R193C; M271del
Prostate cancer	P139Q; A30T; R187H
Renal cancer	E149 *
Sarcoma	R187L; K35Rfs * 11; K35Q
Skin cancer	P89S

**Table 2 cells-12-01895-t002:** Overview of the dysregulation of APE1 observed in different tumors. For each cancer type, the APE1 expression, diagnostic value, and localization are described and complemented by relevant references. n.d.: not defined; OS: overall survival; DFS: disease-free survival; PFS: progression-free survival; RFS: relapse-free survival; acAPE1: acetylated APE1.

Cancer	Expression	Diagnostic Value	Localization	Refs
Bladder cancer (Bca)	Protein overexpression, associated with poor survival and invasion.	Serum and urine levels as a diagnostic biomarker.	In non-invasive, low-grade tumors, localization is mainly in the nucleus. In invasive, high-grade tumors, both nuclear and cytoplasmic localization.	[38,39,40,41,42,43,44]
Breast cancer	Conflicting data on protein expression: in some cases, overexpression is associated with a malignant phenotype and an unfavorable prognosis; in other cases, low APE1 is associated with an aggressive triple-negative phenotype; Deregulation of acAPE1.	n.d.	Nuclear localization.	[45,46,47,48,49,50]
Cervical cancer	High protein expression is associated with lymph node metastasis, EMT, and decreasing radiosensitivity.	n.d.	Moderate and heterogeneous nuclear staining; Radioresistant cervical cancer cell lines show higher levels of cytoplasmic APE1 and lower levels of nuclear proteins.	[51,52,53,54]
Colorectal cancer (CRC)	Protein overexpression in cancer tissues is increasing from benign to malignant forms; Overexpression of acAPE1 in tumor tissues is positively correlated with 5-FU resistance; Presence of both the full-length and the N∆33 proteins;	Serum APE1-autoantibody levels as a diagnostic biomarker.	Nuclear and cytoplasmic localization; The acetylated form accumulates in the nucleus.	[55,56,57,58,59,60,61,62]
Cutaneous Squamous Cell carcinoma (cSCC)	Protein overexpression in tumor tissues, which promotes cell proliferation and migration by EMT.	n.d.	n.d.	[63]
Gastric carcinoma	mRNA and protein upregulation are correlated with lymph node metastasis, depth of invasion, and poor prognosis.	Serum levels as a diagnostic biomarker for metastasis prediction.	Nuclear and cytoplasmic localization.	[64,65,66,67]
Glioma	Conflicting data are available in some cases, APE1 overexpression is shown, whereas in other cases, low mRNA and protein expression is associated with poor OS; APE1 expression increases following treatment and recurrence.	n.d.	Nuclear localization.	[68,69,70,71,72,73]
Head and Neck Squamous Cell carcinoma (HNSCC)	In oral SCC (oSCC), protein overexpression, correlated with nodal status and lymph node invasion, shorter OS and DFS; In laryngeal SCC (LSCC): protein overexpression in cancer tissues; In sino-nasal SCC (sSCC) and SCC with inverted papilloma (SCCwIP): protein overexpression in cancer tissues; In lip SCC (lSCC), protein overexpression in cancer tissues.	In oSCC: serum levels are used as a diagnostic biomarker, with high levels correlated with late TNM stages, lymph node metastasis, and worse pathologic differentiation.	oSCC: mainly nuclear localization with a weak cytoplasmic expression; In sSCC and SCCwIP: vivid nuclear localization, associated with metastasis; higher cytoplasmic staining in sSCC, associated with T-stage and histological grade; *In lSCC:* nuclear localization.	[74,75,76,77,78,79,80]
Liver cancer	mRNA and protein overexpression, correlated with poor survival and cancer aggressiveness; Presence of both the full-length and the N∆33 proteins.	Serum levels as a diagnostic biomarker.	Strong nuclear and cytoplasmic positivity, with higher cytosol expression in poorly differentiated tumors; In lower-grade tumors, cytoplasmic positivity is associated with mitochondrial accumulation.	[81,82,83,84,85,86]
Lung cancer	mRNA and protein overexpression in NSCLC (non-small-cell lung cancer), are associated with linfonodal metastasis and EMT promotion; Increase in APE1 expression after cisplatin treatment; High levels of acAPE1; Presence of both the full length and N∆33 proteins.	High post-treatment serum levels are associated with lower OS.	Nuclear and cytoplasmic staining: higher cytoplasmic localization correlates with poor prognosis; acAPE1 is strictly nuclear.	[55,87,88,89,90,91,92,93,94,95,96]
Melanoma	mRNA and protein overexpression are associated with vascular invasion, a high mitotic rate, lower response to therapy, and a poor prognosis.	n.d.	Nuclear localization.	[97,98,99,100]
Oesophageal carcinoma (EAC)	Protein overexpression in tumor tissues is associated with worse OS.	n.d.	Mainly nuclear localization.	[101,102,103,104,105,106]
Osteosarcoma	Protein overexpression in cancer samples is associated with poor prognosis and cisplatin resistance.	n.d.	Mostly nuclear localization and variable cytoplasmic staining; High cytoplasmic localization correlates with poor response to cisplatin therapy.	[107,108,109,110,111,112,113]
Ovarian cancer	Protein overexpression in tumor tissues is associated with advanced stages, platinum resistance, poor chemosensitivity, decreased OS, and lymph node metastasis.	n.d.	Strong nuclear and cytoplasmic localization, heterogeneous between different histological subtypes; Cytoplasmic localization increases from well-to-poorly differentiated tumors, and it is higher in advanced stages; Cytoplasmic localization is associated with lower PFS time and decreased OS.	[101,114,115,116,117,118,119,120,121]
Pancreatic adenocarcinoma (PDAC)	Protein overexpression in tumor tissues is associated with poor prognosis and tumor aggressiveness; Elevated levels of acAPE1; Presence of both the full-length and the N∆33 proteins.	n.d.	Strong nuclear staining in tumor tissues, with cytosol staining only in advanced stages.	[55,87,101,122,123,124]
Prostate carcinoma (PCa)	Protein overexpression in cancer samples.	n.d.	Nuclear and cytoplasmic localization.	[125,126]
Salivary gland carcinoma	Protein overexpression in tumor tissues increases dependence on malignant transformation and is correlated with lymph node metastasis and invasive growth; Higher protein levels in smaller tumors.	n.d.	Mainly nuclear staining, with nuclear and cytosolic staining in some malignant forms.	[127,128]

## Data Availability

No new data were created with this paper.

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
