# Peer review of "Revisiting Two Decades of Research Focused on Targeting APE1 for Cancer Therapy: The Pros and Cons"

_cells, 2023, doi:10.3390/cells12141895_

Round 1
Reviewer 1 Report
AP endonuclease 1 (APE1) is a multifunctional protein critical for genome integrity. In Malfatti et al. manuscript, the authors aimed to provide an updated perspective on how to target APE1 for cancer therapy. After a short introduction, they provide updates on APE1 polymorphisms and differential expression of APE1 in different cancer types. Next, the authors discussed on how to target APE1 endonuclease activity, APE1 redox function, and in combination for cancer therapy. In addition, the authors discussed new perspectives on APE1 functions in miRNA processing and secreted APE1 as a prognostic biomarker.
Overall, the authors provide an updated perspective on APE1 in cancer therapy. However, below major and minor concerns should be addressed before consideration of publication.
Major concerns:
1) In section 2 relevant to APE1 differential expression in different cancer types, the authors provided a laundry list of altered APE1 expression in cancers; however, this version of analysis lacks sort of further analysis on how and why altered APE1 expression are differentially associated with cancer development and metastasis. Is APE1 a driver or passenger of cancer development?
2) In Section 3.1, the authors summarized the studies of APE1 inhibitors targeting endonuclease activity. However, recent studies have shown that AR03 and APE1 Inhibitor III also inhibit its exonuclease activity. The authors should include these new developments in targeting APE1 inhibitors with appropriate references.
3) In recent literature, APE1 is also identified as a critical regulator in the ATR-mediated DNA damage response. At least a short paragraph of discussion on how targeting APE1 function in ATR for cancer therapy should be included in this updated perspective.
Minor concerns:
1) Line 605=608: “On these lines, specific inhibitors of APE1-onco-miR interaction should be explored as potentially novel anticancer strategies for personalized medicine of cancer abs’ on APE1-miR-specific signatures. Work is ongoing in our Lab along these new lines.” What is “cancer abs’ on APE1-specific signatures? The last sentence should be removed.
2) In Line 586-588: “iii) is involved in EVs-based sorting of specific miRNA through interaction with hnRNPA2/B1 (in preparation”. If this study is not published yet, this statement should be removed.
3) In Line 597-599: “Interesting, bioinformatics analysis, demonstrated that these miRs possess a strong propensity to contain the RG4 structures in their immature or mature forms (unpublished data).” If this study is not published yet, this statement should be removed.
Author Response
Reviewer: 1
Comments and Suggestions for Authors:
AP endonuclease 1 (APE1) is a multifunctional protein critical for genome integrity. In Malfatti et al. manuscript, the authors aimed to provide an updated perspective on how to target APE1 for cancer therapy. After a short introduction, they provide updates on APE1 polymorphisms and differential expression of APE1 in different cancer types. Next, the authors discussed on how to target APE1 endonuclease activity, APE1 redox function, and in combination for cancer therapy. In addition, the authors discussed new perspectives on APE1 functions in miRNA processing and secreted APE1 as a prognostic biomarker.
Overall, the authors provide an updated perspective on APE1 in cancer therapy. However, below major and minor concerns should be addressed before consideration of publication.
We thank this Reviewer for the positive comments on our work.
Major concerns:
Q1) In section 2 relevant to APE1 differential expression in different cancer types, the authors provided a laundry list of altered APE1 expression in cancers; however, this version of analysis lacks sort of further analysis on how and why altered APE1 expression are differentially associated with cancer development and metastasis. Is APE1 a driver or passenger of cancer development?
A1) We thank the Reviewer because we appreciated this comment since highlights an interesting open point regarding the impact of APE1 on tumorigenesis. To date, despite the consistent amount of literature associating APE1 overexpression with cancer cell proliferation and chemoresistance mechanisms, there is still poor understanding of the real role played by APE1 genetic alterations in tumor onset and progression. For these reasons, it is actually difficult to say whether APE1 mutations may acts as drivers or passengers in the tumorigenic process. Therefore, we only reported in this review some hypotheses about the link between its functional alterations and the clinical observations in cancer cell models or tumor patients. However, we believe that these aspects may represent a hot topic in APE1 research field to open new perspectives for developing new specific inhibitory strategies. To highlight all these points in this new revised version of the manuscript, we reported the following paragraphs at lines 73-75, page 2: “It must be clearly stated, however, that, up to now, there is not a clear evidence for a driver or passenger function of neither the above mentioned alterations in the tumorigenic processes.” and at lines 299-320, page 13: “Therefore, in general, APE1 is significantly overexpressed in different kinds of cancers and subcellular distribution may significantly change depending on the specific tissue and tumoral stage, but in which way the overexpression and localization of APE1 in tumors is causally responsible for cancer onset and development, aggressiveness, and invasion is still debated. Currently, knowledge about the role played by APE1 polymorphic variants in cancer onset and progression is still unknown as well as the possible driver or passenger functions of APE1 mutations in cancer tumorigenesis. As above mentioned, the most accredited hypothesis regards the increased expression of APE1 in tumoral cells , as they acquire a proliferative- and chemoresistant- phenotype. Several studies have proposed that the up-regulation of APE1, as well that of other BER enzymes, may underly pro-survival mechanisms adopted by tumors to efficiently repair DNA damage thus contributing to the onset of resistance mechanisms. Although the main function of APE1 is attributable to its endoribonuclease activity, it is believed that APE1 overexpression may also contribute to tumorigenesis through an increased activity as a redox activator of several TFs, such as NF-kB, thus leading to an increase in tumor proliferation and survival, and affecting tumor microenvironment. We do believe that additional dysregulated functions of APE1, including dysregulation of RNA and miR metabolism and regulation of G4-structures containing promoter genes, could play an essential role in cancer development, although more detailed investigations are needed.
In conclusion, further analysis on how and why altered APE1 expression are differentially associated with cancer development and metastasis should be a central aim of further study in the next future.”
Q2) In Section 3.1, the authors summarized the studies of APE1 inhibitors targeting endonuclease activity. However, recent studies have shown that AR03 and APE1 Inhibitor III also inhibit its exonuclease activity. The authors should include these new developments in targeting APE1 inhibitors with appropriate references.
A2) We thank the Reviewer for her/his suggestion. In this new revised version of the manuscript, we included the recent suggested studies at page 17, lines 417-420: “Furthermore, very recently, AR03 was demonstrated to inhibit the exonuclease activity of APE1 in SSB-induced ATR-Chk1 DDR pathway in human bone osteosarcoma U2OS cells, MDA-MB-231 and PANC1 [5,20].”, and page 18, lines 454-457: “This compound was further used and distributed by the sellers as one of the most promising APE1 inhibitors for both its endonuclease and exonuclease activity, however, also this compound has not significantly advanced beyond in vitro studies.”. We included, as suggested, appropriate references that are here listed:
- Lin, Y.; Raj, J.; Li, J.; Ha, A.; Hossain, M.A.; Richardson, C.; Mukherjee, P.; Yan, S. APE1 Senses DNA Single-Strand Breaks for Repair and Signaling. Nucleic Acids Research 2020, 48, 1925–1940, doi:10.1093/nar/gkz1175.
- Li, J.; Zhao, H.; McMahon, A.; Yan, S. APE1 Assembles Biomolecular Condensates to Promote the ATR–Chk1 DNA Damage Response in Nucleolus. Nucleic Acids Research 2022, 50, 10503–10525, doi:10.1093/nar/gkac853.
Q3) In recent literature, APE1 is also identified as a critical regulator in the ATR- mediated DNA damage response. At least a short paragraph of discussion on how targeting APE1 function in ATR for cancer therapy should be included in this updated perspective.
A3) We thank the Reviewer for her/his suggestion. In this new revised version of the manuscript, we dedicated a short paragraph in the Introduction section to the description of the new APE1 function in the ATR/Chk1- mediated DDR response (page 1, lines 34-39): “Recent data have demonstrated that SSBs are also sensed by APE1 to initiate 3’-5’ SSB end resection and to promote the ATR/Chk1-mediated DNA damage response (DDR) activation [5]. Indeed, through its exonuclease activity, APE1 generates a short ssDNA gap, that via PCNA and APE2, becomes a longer stretch of ssDNA coated by RPA that leads to the assembly of the ATR/Chk1 DDR complex [5,6].” Moreover, we emphasized that targeting this new function can be an interesting direction that should be pursued including a short paragraph in the discussion section at page 23, lines 702-706 that sounds like: “The discovery of novel functions of APE1 is constantly evolving. As mentioned in the introduction, the ability of APE1 to recognize and process SSBs through its 3’-5’ exonuclease activity [5,6] could represent an interesting target for developing new inhibitors specifically direct versus this APE1 function and able to inhibit in ultimum the promotion of the ATR/Chk1- mediated DDR activation.”. We included appropriate references that are here listed:
- Lin, Y.; Raj, J.; Li, J.; Ha, A.; Hossain, M.A.; Richardson, C.; Mukherjee, P.; Yan, S. APE1 Senses DNA Single-Strand Breaks for Repair and Signaling. Nucleic Acids Research 2020, 48, 1925–1940, doi:10.1093/nar/gkz1175.
- Lin, Y.; Li, J.; Zhao, H.; McMahon, A.; McGhee, K.; Yan, S. APE1 Recruits ATRIP to SsDNA in an RPA-Dependent and -Independent Manner to Promote the ATR DNA Damage Response. eLife 2023, 12, e82324, doi:10.7554/eLife.82324.
Minor concerns:
Q1) Line 605=608: “On these lines, specific inhibitors of APE1-onco-miR interaction should be explored as potentially novel anticancer strategies for personalized medicine of cancer abs’ on APE1-miR-specific signatures. Work is ongoing in our Lab along these new lines.” What is “cancer abs’ on APE1-specific signatures? The last sentence should be removed.
A1) In accordance with the Reviewer, we deleted the entire statement.
Q2) In Line 586-588: “iii) is involved in EVs-based sorting of specific miRNA through interaction with hnRNPA2/B1 (in preparation”. If this study is not published yet, this statement should be removed.
A2) Although the manuscript has been presently under evaluation for publication, in accordance with the Reviewer, we deleted the entire statement.
Q3) In Line 597-599: “Interesting, bioinformatics analysis, demonstrated that these miRs possess a strong propensity to contain the RG4 structures in their immature or mature forms (unpublished data).” If this study is not published yet, this statement should be removed.
A3) In accordance with the Reviewer, we deleted the entire statement.

Reviewer 2 Report
In this report, the authors nicely summarize the status of reports on the AP endonuclease protein APE1, also called APEX1. While they emphasize work from their own group, they provide a well referenced overall view of APE1 relevant studies, particularly as it relates to its role in cancer and the development and status of APE1 inhibitors.
Overall, this is a well outlined and written report. However, there are a few minor concerns:
1) All the Greek symbols (beta, gamma etc) come through as @ symbols.
2) The authors should get advice with English language editing as word usage, phrases and sentence structure can significantly be improved.
The authors should get advice with English language editing as word usage, phrases and sentence structure can significantly be improved.
Author Response
Reviewer: 2
Comments and Suggestions for Authors:
In this report, the authors nicely summarize the status of reports on the AP endonuclease protein APE1, also called APEX1. While they emphasize work from their own group, they provide a well referenced overall view of APE1 relevant studies, particularly as it relates to its role in cancer and the development and status of APE1 inhibitors.
Overall, this is a well outlined and written report. However, there are a few minor concerns:
We thank this Reviewer for the positive comments on our work.
Q1) All the Greek symbols (beta, gamma etc) come through as @ symbols.
A1) We thank the Reviewer for her/his comment. We think that this issue derived from a problem during the upload of the file in which some formatting features were lost. In any case, we made sure that any symbol is now correctly reported in this revised submission.
Q2) The authors should get advice with English language editing as word usage, phrases and sentence structure can significantly be improved.
A2) We thank this Reviewer for her/his suggestion that allowed us to extensively improve the editing in this new revised version of the manuscript.
